Decrease in walking speed increases hip moment impulse in the frontal plane during the stance phase

Inai Takuma hwd17001@nuhw.ac.jp
Takabayashi Tomoya
Edama Mutsuaki
Kubo Masayoshi
Institute for Human Movement and Medical Sciences, Niigata University of Health and Welfare , Niigata City , Japan
Fuller Joel
Electronic publication date: 2019 Nov 19
Publication date: 2019
Volume: 7
Electronic Location ID: e8110
Received 2019 Jul 9; Accepted 2019 Oct 28
Copyright: ©2019 Inai et al.
Copyright year: 2019
Copyright holder: Inai et al.
License: This is an open access article distributed under the terms of the Creative Commons Attribution License, which permits unrestricted use, distribution, reproduction and adaptation in any medium and for any purpose provided that it is properly attributed. For attribution, the original author(s), title, publication source (PeerJ) and either DOI or URL of the article must be cited.
License URL: https://creativecommons.org/licenses/by/4.0/

Keywords: Hip, Impulse, Walking, Speed

Funding: The authors received no funding for this work.

==============================
Background

Increased daily cumulative hip moment in the frontal plane (i.e., the product of hip moment impulse in the frontal plane during the stance phase and mean steps per day) is a risk factor for progression of hip osteoarthritis. Although hip osteoarthritis generally causes a decrease in the walking speed, its effect on hip moment impulse in the frontal plane is unclear. The purpose of this study was to examine the relationship between decrease in walking speed and hip moment impulse in the frontal plane.

Methods

We used a public dataset of treadmill walking in 17 older adults (mean (SD) age: 63.2 (8.0) years). The subjects walked on the treadmill for 30 s under five conditions: (1) 40% of comfortable non-dimensional speed (CNDS), (2) 55% CNDS, (3) 70% CNDS, (4) 85% CNDS, and (5) 100% CNDS. The hip moment impulse in the frontal plane non-normalized (or normalized) to step length (Nm s/kg [or Nm s/(kg m)]) for each condition was calculated. Furthermore, the relationship between walking speed and hip moment impulse in the frontal plane non-normalized (or normalized) to step length was examined using regression analysis based on a previous study.

Results

A decrease in non-dimensional speed (i.e., walking speed) significantly increased the non-normalized (or normalized) hip moment impulse in the frontal plane during the stance phase. The relationship between walking speed and non-normalized (or normalized) hip moment impulse in the frontal plane was fitted by a second-order polynomial.

Discussion

This study revealed that a decrease in walking speed increased the non-normalized (or normalized) hip moment impulse in the frontal plane in healthy older adults. This finding is useful for understanding the relationship between walking speed and hip moment impulse in the frontal plane and suggests that a decrease in walking speed may actually increase the daily cumulative hip moment in the frontal plane of patients with hip osteoarthritis.

Introduction

Abnormal mechanical loading can lead to the onset and progression of osteoarthritis (DeFrate et al., 2019), with changes to the articular cartilage surface, including irregularities, thinning, and defects in patients with hip osteoarthritis (Ahedi et al., 2016). Previous studies show that hip osteoarthritis causes decreased muscle strength (Loureiro, Mills & Barrett, 2013), decreased range of motion (Steultjens et al., 2000), decreased ability to perform activities of daily living (e.g., walking and stair climbing (Pua et al., 2009)), and pain in the hip joint (Kumar et al., 2013). Therefore, it is important to prevent the progression of hip osteoarthritis to enable patients to retain their physical function (i.e., muscle strength and range of motion) and ability to perform activities of daily living.

Recently, Tateuchi et al. (2017) examined the relationships between radiographic progression (i.e., the hip joint space width) and various gait parameters (e.g., peak external hip moments) during the stance phase and found that a higher daily cumulative hip moment in the frontal plane (i.e., the product of hip moment impulse in the frontal plane and number of steps taken per day) predicted radiographic progression of hip osteoarthritis over a 12-month period. In other words, an excessive increase in the hip moment impulse in the frontal plane or the number of steps taken per day is a risk factor for hip osteoarthritis and a decrease in these parameters may prevent its progression.

Many previous studies (Constantinou et al., 2017; Foucher, 2017; Meyer et al., 2018; Wesseling et al., 2018; Diamond et al., 2018) have reported that walking speed of patients with hip osteoarthritis is significantly lower than that of the controls. It has been reported that a change in the walking speed causes changes in kinetics (e.g., the peak hip adduction moment (Rutherford & Hubley-Kozey, 2009; Ardestani et al., 2016; Chehab, Andriacchi & Favre, 2017)); therefore, a decrease in walking speed may affect the hip moment impulse in the frontal plane. However, these previous studies (Rutherford & Hubley-Kozey, 2009; Ardestani et al., 2016; Chehab, Andriacchi & Favre, 2017) did not examine the effect of a decrease in walking speed on the hip moment impulse in the frontal plane during gait. Further, although a recent systematic review (Inai et al., 2018) analyzed several previous studies that examined the hip moment impulse in the frontal plane during gait, the effect of a decrease in the walking speed on the hip moment impulse in the frontal plane has not been reported.

Clarifying the effect of a decrease in walking speed on hip moment impulse in the frontal plane may help understand the gait pattern with high or low hip moment impulse in the frontal plane. Furthermore, this knowledge may provide an important insight to delaying the progression of hip osteoarthritis in the future. Therefore, in the present study, we aimed (1) to examine the effect of a decrease in walking speed on the hip moment in the frontal plane during the stance phase and (2) to clarify the relationship between decrease in walking speed and hip moment impulse in the frontal plane using nonlinear regression analysis. We proposed two hypotheses as follows: (1) decrease in walking speed would increase the hip moment impulse in the frontal plane and (2) the relationship between decrease in walking speed and hip moment impulse in the frontal plane would be nonlinear.

Materials & Methods

A public dataset of treadmill walking

We used a public dataset of treadmill walking provided in a previous study (Fukuchi, Fukuchi & Duarte, 2018). Forty-two volunteers participated in the previous study (24 young adults and 18 older adults). According to the previous study (Fukuchi, Fukuchi & Duarte, 2018), each participant read and signed a consent form that had previously been approved by the ethics committee of the Federal University of ABC prior to the experiment (Approval number: 53063315.7.0000.5594).

A dual-belt instrumented treadmill (FIT, USA) and motion-capture system (Cortex 6.0, USA) were used to record the treadmill walking for each subject. The marker trajectories and ground reaction forces were acquired at 150 Hz and 300 Hz, respectively (Fukuchi, Fukuchi & Duarte, 2018). Twenty-two reflective markers during treadmill walking were captured by the motion-capture system (please see Fukuchi, Fukuchi & Duarte (2018) for details).

In the previous study (Fukuchi, Fukuchi & Duarte, 2018), the task was walking for 30 s on the treadmill under eight conditions: (1) 40% of comfortable non-dimensional speed (Hof, 1996) (CNDS), (2) 55% CNDS, (3) 70% CNDS, (4) 85% CNDS, (5) 100% CNDS, (6) 115% CNDS, (7)130% CNDS, and (8) 145% CNDS. In addition, the relationship between non-dimensional speed (Froude number) and walking speed was calculated as follows (Hof, 1996): v∗=v∕gl0,

where v∗ is non-dimensional speed (Froude number), v is walking speed, g is gravitational acceleration (9.81 m/s2), and l0 is leg length.

They used Visual 3D software version 6.00.33 (C-motion Inc.) to calculate the kinetics (hip, knee, and ankle joint moments of both legs) of treadmill walking for each condition (Fukuchi, Fukuchi & Duarte, 2018). Furthermore, the time-normalized average curves of the kinetics for each treadmill walking condition and subject were calculated.

Data analysis in the present study

Because a lot of patients with hip osteoarthritis are older adults according to a previous systematic review (Diamond et al., 2018), we used the public dataset of treadmill walking on older adults (n = 17 (10 men and seven women), mean (SD) age: 63.2 (8.0) years, height: 1.63 (0.1) m, and body mass: 66.5 (10.2) kg). One older adult (Subject number: 41 (Fukuchi, Fukuchi & Duarte, 2018)) was excluded because we found technical errors in the values of vertical ground reaction forces for each leg (File name: WBDS41walkT01grf.txt). Furthermore, to examine the effect of a decrease in walking speed on hip moment impulse in the frontal plane during the stance phase, we used the dataset of treadmill walking under five conditions: (1) 40% CNDS, (2) 55% CNDS, (3) 70% CNDS, (4) 85% CNDS, and (5) 100% CNDS.

Initially, ground reaction forces and reflective marker trajectories of all trials were filtered using a fourth-order, zero-lag Butterworth low-pass filter with a 10-Hz and 6-Hz cut-off frequency, respectively. Using a vertical ground reaction force of a trial during treadmill walking (30 s), all stance times (s) from right (left) heel contact to right (left) toe off were calculated and averaged. Further, all gait cycle times (s) from right (left) heel contact to next right (left) heel contact were calculated and averaged. Using all gait cycle times of a trial, cadences (steps/min) were calculated and averaged. Step lengths (m) at all heel contacts were calculated (i.e., anterior/posterior distance from right heel marker to left heel maker) and averaged. Walking speeds (m/s) of all the subjects were obtained from (WBDSinfo.xlsx) the previous study (Fukuchi, Fukuchi & Duarte, 2018).

A previous study (Tateuchi et al., 2017) proposed that the daily cumulative hip moment in the frontal plane (Nm s) was the product of hip moment impulse in the frontal plane (Nm s) and the number of steps taken per day (steps/day), and this study analyzed the hip moment impulse in the frontal plane during the stance phase. Therefore, the present study also analyzed stance phase (i.e., from right (left) heel contact to right (left) toe-off). Because the public dataset used in the study includes time-normalized curves, the data of kinetics during the stance phase were extracted using gait cycle time and stance time for each subject. Thereafter, the hip moment impulse in the frontal plane during the stance phase was calculated by time-integrating the hip abduction and adduction moments during the same phase (i.e., the area under the external hip abduction and adduction moments (Inai et al., 2018)).

In the present study, we define the unit of “hip moment impulse in the frontal plane” as Nm s. Further, we define the unit of “non-normalized hip moment impulse in the frontal plane” as Nm  s/kg (with “non-normalized” meaning that hip moment impulse in the frontal plane is not normalized to step length). A difference in step length affects daily cumulative hip moment impulse in the frontal plane as shown in Fig. 1, even when non-normalized hip moment impulse in the frontal plane and walking distance per day are constant. Based on a previous study (Tateuchi et al., 2017), the equation of daily cumulative hip moment in the frontal plane is shown as follows: D=Inon-normalizedBS2L,

where D is daily cumulative hip moment in the frontal plane (Nm s), Inon-normalized is non-normalized hip moment impulse in the frontal plane (Nm s/kg), B is body mass (kg), S is walking distance per day (km), and L is step length (m). We then changed the above equation as follows: Inormalized=Inon-normalizedL=2DBS,

where Inormalized is “normalized hip moment impulse in the frontal plane (Nm s/(kg m)) (i.e., “normalized” here meaning that the hip moment impulse in the frontal plane is normalized to step length). Body mass (B, in kg) and walking distance per day of a person (S, in km) are constant. From these equations, we can understand a change in daily cumulative hip moment impulse in the frontal plane indirectly by using normalized hip moment impulse in the frontal plane Inormalized. Therefore, we also used the index named normalized hip moment impulse in the frontal plane Inormalized.

Figure 1 Effect of step length on daily cumulative hip moment in the frontal plane.

(A) The step length is 0.50 m. (B) The step length is 0.45 m. The non-normalized hip moment impulse in the frontal plane, body mass of the subject, and walking distance per day are assumed as 0.30 Nm s/kg, 65 kg, and 4.0 km, respectively. Tateuchi et al. (2017) defined the unit of daily cumulative hip moment in the frontal plane as Nm s. A decrease in step length increases the daily cumulative hip moment in the frontal plane. This figure has been created using OpenSim 4.0 (Delp et al., 2007).

Furthermore, to examine the validity of hip adduction moment during the stance phase, first peak hip adduction moment in the first half of the stance phase and hip adduction moment at 50% of the stance phase were also calculated. Data analysis was conducted using Scilab (Scilab Enterprises, France).

Statistical analysis

To compare gait characteristics (stance time, walking speed, cadence, and step length) and kinetics (non-normalized (or normalized) hip moment impulse in the frontal plane, first peak hip adduction moment, and hip adduction moment at 50% of the stance phase) among five conditions, multiple comparisons (the Bonferroni method) were performed. First, the Shapiro–Wilk test was used to determine whether all variables followed a normal distribution. According to the results of normality, the paired t-test or Wilcoxon signed-rank test with Bonferroni corrections was used. All obtained p-values were adjusted to set the level of significance at 0.05.

To clarify the relationship between walking speed and non-normalized (or normalized) hip moment impulse in the frontal plane, a regression analysis by Levenberg–Marquardt method was performed. A previous study (Fukuchi & Duarte, 2019) has reported that a relationship between walking speed and kinetics (e.g., joint moment) may be nonlinear. Therefore, we also examined the statistical significance of the coefficient a of the second-order polynomial regression (y = ax2 + bx + c; y is non-normalized (or normalized) hip moment impulse in the frontal plane, and x is walking speed). If this coefficient was significant, a second-order polynomial was employed (Fukuchi & Duarte, 2019) (if this coefficient was not significant, a first-order polynomial was employed). All statistics procedures (multiple comparisons and regression analysis) were performed using the R language (R Development Core Team, Vienna, Austria).

Results

Figure 2A shows the effect of a decrease in non-dimensional speed (i.e., walking speed) on normalized hip moment impulse in the frontal plane. A decrease in walking speed significantly increased normalized hip moment impulse in the frontal plane. The mean (SD) normalized hip moment impulses in the frontal plane in 40%, 55%, 70%, 85%, and 100% CNDS were 1.33 (0.31), 0.97 (0.22), 0.78 (0.16), 0.64 (0.13), and 0.55 (0.11) Nm s/(kg m), respectively. The mean ratios of normalized hip moment impulse in the frontal plane in 40%, 55%, 70%, and 85% CNDS to that in 100% CNDS were 2.4, 1.8, 1.4, and 1.2, respectively. Figure 2B shows the effect of a decrease in non-dimensional speed on non-normalized hip moment impulse in the frontal plane. A decrease in walking speed significantly increased non-normalized hip moment impulse in the frontal plane. The mean (SD) non-normalized hip moment impulses in the frontal plane in 40%, 55%, 70%, 85%, and 100% CNDS were 0.47 (0.10), 0.41 (0.09), 0.36 (0.07), 0.33 (0.06), and 0.31 (0.05) Nm s/kg, respectively. The mean ratios of non-normalized hip moment impulse in the frontal plane in 40%, 55%, 70%, and 85% CNDS to that in 100% CNDS were 1.5, 1.3, 1.2, and 1.1, respectively.

Figure 2 Normalized and non-normalized hip moment impulses in the frontal plane in all conditions.

(A) A decrease in non-dimensional speed increases the normalized hip moment impulse in the frontal plane. (B) A decrease in non-dimensional speed increases the non-normalized hip moment impulse in the frontal plane. *, p < 0.05.

Figures 3A and 3B show the relationships between walking speed and normalized (or non-normalized) hip moment impulse in the frontal plane. The statistical significances of the coefficients of the second-order polynomials were examined, and these coefficients were significant (Figs. 3A and 3B).

Figure 3 Relationship between walking speed and normalized (or non-normalized) hip moment impulse in the frontal plane.

(A) y1 is normalized hip moment impulse in the frontal plane (Nm s/(kg m)). (B) y2 is non-normalized hip moment impulse in the frontal plane (Nm s/kg). x is walking speed (m/s).

Figure 4 shows waveforms of hip adduction moments of all conditions. Furthermore, Figure 5A shows the effect of a decrease in non-dimensional speed (i.e., walking speed) on first peak hip adduction moment, and a decrease in non-dimensional speed significantly decreased first peak hip adduction moment (100% CNDS vs. 40, 55, 70, and 85% CNDS). The mean (SD) first peak hip adduction moment in 40%, 55%, 70%, 85%, and 100% CNDS were 0.77 (0.11), 0.75 (0.12), 0.77 (0.11), 0.79 (0.11), and 0.84 (0.11) Nm/kg, respectively. Figure 5B indicates that a decrease in non-dimensional speed increases hip adduction moment at 50% of the stance phase. The mean (SD) hip adduction moment at 50% of the stance phase in 40%, 55%, 70%, 85%, and 100% CNDS were 0.72 (0.10), 0.68 (0.10), 0.65 (0.09), 0.60 (0.10), and 0.55 (0.09) Nm/kg, respectively.

Figure 4 Averaged waveforms of hip moments in the frontal plane in all conditions.

In 70%, 85%, and 100% CNDS, bimodal curves of hip adduction moments are observed.

Figure 5 (A) First peak hip adduction moment in first half of the stance phase and (B) hip adduction moment at 50% of the stance phase in all conditions.

(A) A decrease in non-dimensional speed decreases first peak hip adduction moment. (B) A decrease in non-dimensional speed increases hip adduction moment at 50% of the stance phase. *, p < 0.05.

Table 1 shows the results of the gait characteristics. A decrease in non-dimensional speed significantly increased stance time. Further, a decrease in non-dimensional speed significantly decreased walking speed, cadence, and step length.

Table 1 Gait characteristics in all conditions.

	40% CNDS	55% CNDS	70% CNDS	85% CNDS	100% CNDS	
Gait characteristics											
Stance time, s	0.95	(0.15)	0.80	(0.10)*	0.70	(0.08)*,†	0.64	(0.06)*,†,‡	0.59	(0.05)*,†,‡,§	
Walking speed, m/s	0.50	(0.07)	0.69	(0.10)*	0.87	(0.13)*,†	1.06	(0.15)*,†,‡	1.25	(0.18)*,†,‡,§	
Cadence, steps/min	83.2	(12.5)	95.0	(11.8)*	107.2	(12.2)*,†	115.5	(11.2)*,†,‡	124.3	(10.4)*,†,‡,§	
Step length, m	0.36	(0.04)	0.42	(0.04)*	0.47	(0.06)*,†	0.52	(0.06)*,†,‡	0.56	(0.07)*,†,‡,§	
Notes.

CNDS comfortable non-dimensional speed

* Significantly difference when compared with 40% CNDS (p < 0.05).

† Significantly difference when compared with 55% CNDS (p < 0.05).

‡ Significantly difference when compared with 70% CNDS (p < 0.05).

§ Significantly difference when compared with 85% CNDS (p < 0.05).

Discussion

In this study, we examined the effect of a decrease in walking speed (i.e., non-dimensional speed) on the normalized (or non-normalized) hip moment impulse in the frontal plane during the stance phase. Furthermore, we clarified the relationship between walking speed and normalized hip moment impulse in the frontal plane. Our main findings were as follows: (1) a decrease in walking speed increased normalized hip moment impulse in the frontal plane (Fig. 2A) and (2) the relationship between walking speed and the normalized hip moment impulse in the frontal plane was nonlinear (a second-order polynomial) (Fig. 3A). Considering these findings, our hypotheses were correct.

As it was previously reported that a higher knee adduction moment impulse is a risk factor for the progression of knee osteoarthritis (Bennell et al., 2011; Chang et al., 2015), many studies have investigated the knee adduction moment impulse during gait (Simic et al., 2010; Gerbrands et al., 2017; Khan, Khan & Usman, 2017; Madden et al., 2017). However, it was recently reported that an excessive increase in the hip moment impulse in the frontal plane is a risk factor for the progression of hip osteoarthritis (Tateuchi et al., 2017). In other words, it is important to understand the gait pattern with high hip moment impulse in the frontal plane during the stance phase to prevent the progression of hip osteoarthritis. However, there are few studies that examined this index (Inai et al., 2018). Moreover, although the effect of walking speed on the peak hip adduction moment during the stance phase has been reported (Ardestani et al., 2016; Chehab, Andriacchi & Favre, 2017), the effect on the normalized (or non-normalized) hip moment impulse in the frontal plane has not been examined. For these reasons, our findings are novel and may prove important for understanding the gait pattern with high normalized (or non-normalized) hip moment impulse in the frontal plane.

The normalized (or non-normalized) hip moment impulse in the frontal plane increased sharply (non-linearly) with a decrease in walking speed (Fig. 2). In the present study, the stance time increased with a decrease in walking speed (Table 1), and these results further validate a previous study (Andriacchi, Ogle & Galante, 1977). The hip moment impulse in the frontal plane is calculated by time-integrating the external hip abduction and adduction moments during the same phase; therefore, a long stance time owing to a decrease in walking speed can cause high hip moment impulse in the frontal plane. From these findings, we consider that the effect of stance time on hip moment impulse in the frontal plane is large. Therefore, to avoid high daily cumulative hip moment in the frontal plane, it may be important not to increase stance time of patients with hip osteoarthritis.

Furthermore, results of the present study suggest that a decrease in walking speed decreases the step length and cadence (Table 1). If total daily walking distance for a person is same (e.g., 4.0 km), a shorter step length will increase the total number of steps. Therefore, to avoid high daily cumulative hip moment in the frontal plane, it may also be important to avoid decreasing step length for patients with hip osteoarthritis.

A previous study by Tateuchi et al. (2017) has reported the value of non-normalized hip moment impulse in the frontal plane during the stance phase as 0.41 (0.13) Nm s/kg (calculated from 22.7 (7.4) Nm s/55.2 kg) at 1.1 m/s. Another study by Tateuchi et al. (2016) reported this value as 0.40 (0.12) Nm s/kg (calculated from 21.9 (6.8) Nm s/54.7 kg) at 1.15 m/s. Alkjær et al. (2006) reported it as 0.37 (0.04) Nm s/kg (calculated from 24.0 (2.7) Nm s/64.7 kg) at 1.25 m/s. In this study, the mean (SD) non-normalized hip moment impulses in the frontal plane (Table 1 and Fig. 2) were 0.33 (0.06) Nm s/kg at 1.06 (0.15) m/s (85% CNDS) and 0.31 (0.05) Nm s/kg at 1.25 (0.18) m/s (100% CNDS), respectively. Therefore, the values of the present study are quantitatively similar to those of previous studies (Alkjær et al., 2006; Tateuchi et al., 2016; Tateuchi et al., 2017), and we consider that the values are reasonable.

Moreover, in the present study, a decrease in walking speed decreased the first peak hip adduction moment in the first half of the stance phase and increased hip adduction moment at 50% of the stance phase (Figs. 5A and 5B). These results are in agreement with those of a previous study (Chehab, Andriacchi & Favre, 2017). Furthermore, a bimodal curve was observed during comfortable walking speed (i.e., 100% CNDS) as shown in Fig. 4, and this result agrees with that of other previous studies (Ardestani et al., 2016; Chehab, Andriacchi & Favre, 2017). For these reasons, we consider that our calculation of kinetics (i.e., hip adduction moment) is optimal.

There are some limitations of this study. First, the subjects included from a public dataset (Fukuchi, Fukuchi & Duarte, 2018) did not have hip osteoarthritis. A previous study (Eitzen et al., 2012) reported that the joint angle and joint moment during gait in patients with hip osteoarthritis were different from those of the control subjects. Although the present study revealed new findings in healthy older adults, it was unclear whether they are applicable to patients with hip osteoarthritis. Second, walking speed is determined by cadence and step length. During walking with slow speed, the strategy of patients with hip osteoarthritis may be different from that of healthy older adults (e.g., high cadence and short step length). Therefore, we consider that the effect of change in step length or cadence on the normalized (or non-normalized) hip moment impulse in the frontal plane should be further examined.

Conclusions

We examined the effect of a decrease in walking speed on the normalized (or non-normalized) hip moment impulse in the frontal plane during the stance phase. Our conclusions were as follows: (1) a decrease in walking speed increased the normalized (or non-normalized) hip moment impulse in the frontal plane and (2) the relationship between walking speed and the normalized (or non-normalized) hip moment impulse in the frontal plane was nonlinear (second-order polynomial). The findings are useful for understanding the relationship between walking speed and normalized (non-normalized) hip moment impulse in the frontal plane.

We would like to thank Editage for English language editing.

Additional Information and Declarations

Competing Interests

Author Contributions

Human Ethics

Data Availability

The authors declare there are no competing interests.

Takuma Inai analyzed the data, contributed reagents/materials/analysis tools, prepared figures and/or tables, authored or reviewed drafts of the paper, approved the final draft.

Tomoya Takabayashi, Mutsuaki Edama and Masayoshi Kubo analyzed the data, contributed reagents/materials/analysis tools, authored or reviewed drafts of the paper, approved the final draft.

The following information was supplied relating to ethical approvals (i.e., approving body and any reference numbers):

The Federal University of ABC ethics committee granted ethical approval to carry out the study within its facilities (CAAE: 53063315.7.0000.5594).

The following information was supplied regarding data availability:

The raw data are available at Figshare: Fukuchi, Claudiane; Fukuchi, Reginaldo; Duarte, Marcos (2018): A public data set of overground and treadmill walking kinematics and kinetics of healthy individuals. figshare. Dataset. https://doi.org/10.6084/m9.figshare.5722711.v4.

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
