# Peer review of "Decrease in walking speed increases hip moment impulse in the frontal plane during the stance phase"

_PeerJ, doi:10.7717/peerj.8110_

## Round 0.1 · original submission · Minor Revisions

The reviewers provided positive feedback about your manuscript and have suggested a small number of minor revisions. Please address all their comments in your revised submission.

·

Basic reporting

no comment

Experimental design

no comment

Validity of the findings

no comment

Additional comments

While I understand why the authors focused on the hip moment at the frontal plane, the manuscript would potentially have a greater impact (contribution to the scientific community) if the same analysis was performed for the knee moment (to confirm previous results from the literature). Also, there is the classic concept of support moment at the sagittal plane (by D. Winter), why not explore that for the frontal plane?

The authors recognize the speculative nature of their rationale regarding the link between their analysis of healthy subjects and the implications to hip osteoarthritis, but I think the last lines of the abstract complete ignores the study limitations and the recommendation "to maintain a general walking speed to delay the progression of hip osteoarthritis" is still too speculative.

Lines 223-227:
"The ratios of normalized hip moment impulse in the frontal plane increased sharply with
a decrease in walking speed (Fig. 2) compared to those of hip moment impulse in the frontal
plane. From this thing, we consider that it important to normalize hip moment impulse in the
frontal plane by step length. Furthermore, we consider that this result suggests the importance of
maintaining normal step length to prevent the progression of hip osteoarthritis."
The justification for using normalized hip moment impulse simply because it amplifies the observed difference between different speeds is not enough (I felt this is the argument in this paragraph). I suggest to develop more this rationale using again the concept of daily cumulative load mentioned earlier.

Reviewer 2 ·

Basic reporting

no comment

Experimental design

no comment

Validity of the findings

no comment

Additional comments

First I would thank authors for the effort they made in this work. This is a well-written paper and may enrich the knowledge about the effect of walking speed and the progression of OA.

I have minor comments that may need to be addressed:
- authors may need to define the non-dimensional speed to make it clearer and to differentiate it from walking speed.
- I think authors need to provide interpretation on the non-linearity of relationship between walking speed and hip moment impulse. and what does that mean?
- line 43: remove the repeated word "surface"
- line 221: do you mean "high" instead of "low" ?
- the reason normalizing the hip moment impulse to step length is not convincing, pointing to the equations in Figure 1 the step length is expected to affect (increase or decrease)the cumulative hip impulse because it is used to calculate the number of steps but I don't think this is the case for normalizing hip moment impulse.
- regarding the equations in figure 1, clarify the reason for multiplying with body mass despite that the impulse is already normalized to body mass?
- line 225: add "is" before "important to normalize...."
- authors may comment on the effect of reduced walking speed on cadence; the slow walking may be achieved with less number of steps and this may reduce the cumulative hip moment impulse

---

## Round 0.2 · accepted · Accept

Thank you for considering the comments from the reviewers and revising your manuscript accordingly.